A new therocephalian (Gorynychus masyutinae gen. et sp. nov.) from the Permian Kotelnich locality, Kirov Region, Russia

Kammerer Christian F. 1 christian.kammerer@naturalsciences.org
Masyutin Vladimir 2
1 North Carolina Museum of Natural Sciences , Raleigh, NC , USA
2 Vyatka Paleontological Museum , Kirov , Russia
Sues Hans-Dieter
Electronic publication date: 2018 Jun 8
Publication date: 2018
Volume: 6
Electronic Location ID: e4933
Received 2018 Mar 15; Accepted 2018 May 17
Copyright: © 2018 Kammerer and Masyutin
Copyright year: 2018
Copyright holder: Kammerer and Masyutin
License: This is an open access article distributed under the terms of the Creative Commons Attribution License, which permits unrestricted use, distribution, reproduction and adaptation in any medium and for any purpose provided that it is properly attributed. For attribution, the original author(s), title, publication source (PeerJ) and either DOI or URL of the article must be cited.
License URL: https://creativecommons.org/licenses/by/4.0/

Keywords: Synapsida, Therapsida, Therocephalia, Permian, Russia, Phylogeny

Funding: Deutsche Forschungsgemeinschaft KA 4133/1-1 This work was supported by a grant from the Deutsche Forschungsgemeinschaft (KA 4133/1-1) to Christian F. Kammerer. The funders had no role in study design, data collection and analysis, decision to publish, or preparation of the manuscript.

==============================
A new therocephalian taxon (Gorynychus masyutinae gen. et sp. nov.) is described based on a nearly complete skull and partial postcranium from the Permian Kotelnich locality of Russia. Gorynychus displays an unusual mixture of primitive (“pristerosaurian”) and derived (eutherocephalian) characters. Primitive features of Gorynychus include extensive dentition on the palatal boss and transverse process of the pterygoid, paired vomers, and a prominent dentary angle; derived features include the absence of the postfrontal. Gorynychus can be distinguished from all other therocephalians by its autapomorphic dental morphology, with roughly denticulated incisors and postcanines. Phylogenetic analysis recovers Gorynychus as a non-lycosuchid, non-scylacosaurid therocephalian situated as sister-taxon to Eutherocephalia. The identification of Gorynychus as the largest predator from Kotelnich indicates that therocephalians acted as apex predators in middle–late Permian transition ecosystems in Russia, corroborating a pattern observed in South African faunas. However, other aspects of the Kotelnich fauna, and Permian Russian tetrapod faunas in general, differ markedly from those of South Africa and suggest that Karoo faunas are not necessarily representative of global patterns.

Introduction

Therocephalians had perhaps the most unusual evolutionary trajectory of the major clades of non-mammalian therapsids. Whereas other non-mammalian therapsid groups remained relatively static in terms of niche occupation through time, therocephalians “reinvented” themselves several times in their history, each time following mass extinctions. The earliest known therocephalians (Lycosuchidae and Scylacosauridae, historically united in the paraphyletic group “Pristerosauria” (Boonstra, 1953)), which are best represented in middle Permian sediments of the Karoo Basin of South Africa, were large-bodied (skull length up to 40 cm) predators (Boonstra, 1969; van den Heever, 1980, 1994). Following the extinction of dinocephalians (including the gigantic, carnivorous anteosaurs) at the end of the Capitanian, therocephalians briefly served as the apex predators of the Karoo during the middle–late Permian transition represented by the Pristerognathus Assemblage Zone (AZ) (Kammerer, 2011; Abdala et al., 2014; Day et al., 2015). However, by the end of the Pristerognathus AZ lycosuchids and scylacosaurids were extinct, and the saber-toothed gorgonopsians had taken over as the dominant large-bodied therapsid predators (Smith, Rubidge & van der Walt, 2012; Kammerer et al., 2015). The surviving late Permian therocephalians all belong to the subclade Eutherocephalia, which were predominantly small-bodied animals (skull length ≤10 cm), many of which were likely insectivores (Mendrez, 1975; Kemp, 1986; Huttenlocker, 2009). A few eutherocephalians re-evolved large size and inferred macropredatory habits by the end of the Permian (e.g., the whaitsiid Theriognathus and the akidnognathid Moschorhinus), but these taxa died out as a result of the end-Permian mass extinction (Moschorhinus survived the main extinction pulse, but disappears from the record shortly thereafter, making it an example of a “dead clade walking”; Jablonski, 2002; Huttenlocker & Botha-Brink, 2013; Huttenlocker, 2014). Remarkably, despite major losses in the end-Permian mass extinction, therocephalians managed to reinvent themselves yet again, with a moderately successful third act as small-bodied herbivores (the Bauriamorpha) in the Early–Middle Triassic (Sigogneau-Russell & Sun, 1981; Abdala et al., 2014) before the clade was finally lost for good.

Basal therocephalians (lycosuchids and scylacosaurids) are the most common tetrapod predators in middle and earliest late Permian deposits in South Africa, with hundreds of known specimens (Smith, Rubidge & van der Walt, 2012) and 55 named species (although this number is clearly oversplit; van den Heever, 1987). By contrast, very few basal therocephalian fossils have been found in comparably-aged Laurasian rocks, despite extensive records of anomodonts, dinocephalians, and parareptiles from the middle Permian of China and Russia (Li, 2001; Ivakhnenko, 2003). No therocephalians have ever been found in middle Permian Chinese rocks. Historically, only a single species of therocephalian (Porosteognathus efremovi, a possible scylacosaurid from the Isheevo locality, Apastovskii District, Tatarstan) was known from the middle Permian of Russia (Vjuschkov, 1955; Ivakhnenko, 2011). Furthermore, Porosteognathus seems to be a minor component of the Isheevo assemblage, which is dominated by dinocephalians and venyukovioid anomodonts (many known from complete skulls and skeletons, whereas Porosteognathus is known only from isolated skull bones).

The earliest Russian assemblage preserving a substantial number of therocephalians is the Kotelnich locality in Kirov Region. Although known since the 1930s as a source of spectacularly-complete pareiasaurs (Hartmann-Weinberg, 1937), therocephalians were not described from Kotelnich until the 1990s (Tatarinov, 1995a, 1995b, 1997, 1999a, 1999b). Now, however, they are the most species-rich tetrapod clade known from the site, with eight named species (Chlynovia serridentatus, Karenites ornamentatus, Kotelcephalon viatkensis, Muchia microdenta, Perplexisaurus foveatus, Scalopodon tenuisfrons, Scalopodontes kotelnichi, and Viatkosuchus sumini), although these may be somewhat oversplit (Ivakhnenko, 2011). The age of the Kotelnich assemblage is somewhat uncertain, with both middle and late Permian ages having been proposed (Tatarinov, 2000; Benton et al., 2012). Currently, an early late Permian age is considered most likely, possibly equivalent with the South African Tropidostoma AZ based on anomodont comparisons (Kurkin, 2011). Benton et al. (2012) instead suggested equivalency between the Kotelnich assemblage and the Pristerognathus AZ. However, the described therocephalian fauna of Kotelnich is composed entirely of eutherocephalians (which other than Viatkosuchus are very small, i.e., <10 cm skull length), not the large scylacosaurids or lycosuchids characteristic of the Pristerognathus AZ in South Africa.

Here we describe a new taxon representing the first large, basal (i.e., non-eutherocephalian) therocephalian from the Kotelnich locality. This species is represented by two specimens and is the largest known predatory therapsid from Kotelnich, indicating therocephalian occupation of apex predator niches in the Northern as well as Southern Hemisphere during the transition between middle and late Permian tetrapod faunas.

Nomenclatural acts

The electronic version of this article in portable document format will represent a published work according to the International Commission on Zoological Nomenclature (ICZN), and hence the new names contained in the electronic version are effectively published under that Code from the electronic edition alone. This published work and the nomenclatural acts it contains have been registered in ZooBank, the online registration system for the ICZN. The ZooBank Life Science Identifiers (LSIDs) can be resolved and the associated information viewed through any standard web browser by appending the LSID to the prefix http://zoobank.org/. The LSID for this publication is: urn:lsid:zoobank.org:pub:CA4D73A1-8FA7-40DD-A464-621AC01421B6. The online version of this work is archived and available from the following digital repositories: PeerJ, PubMed Central, and CLOCKSS.

Geological Context

The Kotelnich locality is a rich, tetrapod-bearing fossil assemblage in the Kirov Region of European Russia. The first fossils found at this locality were two pareiasaur skeletons collected by the hydrogeologist S. G. Kashtanov in 1933 (Kashtanov, 1934). Subsequent expeditions by A. P. Hartmann-Weinberg (in 1935) and Kashtanov (in 1936) recovered additional pareiasaur material (today all considered referable to Deltavjatia rossica; Benton et al., 2012; Tsuji, 2013). Further expeditions by staff of the Paleontological Institute in Moscow occurred between the 1940s–1960s, collecting mostly Deltavjatia specimens (see, e.g., Efremov & Vjuschkov, 1955). Fossil collection at Kotelnich was renewed in the 1990s and has continued to the present day, most recently through the efforts of the Vyatka Paleontological Museum in Kirov. These more recent (1990s–present) excavations have revealed a substantially more diverse fauna than was previously known, adding an array of anomodont, gorgonopsian, and therocephalian therapsids as well as non-pareiasaurian parareptiles to the list of Kotelnich tetrapods (although Deltavjatia remains the numerically dominant taxon; Benton et al., 2012).

Several fossiliferous layers are present at Kotelnich; the lowest red beds represent a lacustrine or floodplain system which famously preserves numerous complete, fully articulated skeletons of the mid-sized pareiasaur D. rossica. This level (the Vanyushonki Member of Coffa (1999)) has also produced the majority of synapsid finds, including spectacular examples such as the complete, articulated specimen of the small gorgonopsian Viatkogorgon ivakhnenkoi (Tatarinov, 1999a) and a block containing 15 skeletons of the arboreal anomodont Suminia getmanovi (Fröbisch & Reisz, 2011). Although most Kotelnich tetrapods are found in the red–brown mudstones at the base of the succession, plants, fish, and highly fragmentary tetrapod remains (primarily isolated teeth) are also present in lenses of later deposition at the top of the section (Benton et al., 2012).

The specimens of the new therocephalian described herein were all found in the lower red beds (Vanyushonki Member). The holotype was discovered in 2008 by I. Shumov, 10.15 m below the marker bed and 413 m upstream from the village of Nizhnyaya Vodskaya. These specimens were mechanically prepared by O. Masyutina and are housed in the collections of the Vyatka Paleontological Museum in Kirov.

Systematic Paleontology

Synapsida Osborn, 1903

Therapsida Broom, 1905

Therocephalia Broom, 1903

Gorynychus gen. nov.

LSID: urn:lsid:zoobank.org:act:CD10EB7C-57C0-45BA-8467-75309411E0DD

Type species: Gorynychus masyutinae sp. nov.

Etymology: Named for the legendary Russian dragon Zmey Gorynych (Змей Горыныч), in reference to the fearsome appearance of this taxon and its status as the largest known predator in the Kotelnich assemblage. Also a play on the English word “gory” (meaning bloody) and the Ancient Greek ὄνῠχος (Latinized “onychus,” meaning claw), in reference to this taxon’s inferred behavior being “red in tooth and claw.”

Diagnosis: As for type and only species.

Gorynychus masyutinae sp. nov.

(Figs. 1–11)

Figure 1 Holotype of Gorynychus masyutinae.

The two blocks (KPM 346 and 347) making up the majority of the holotype shown in articulation. Holotype also includes two incisor teeth (KPM 348 and 349) disarticulated from the skull but found in association (see Figs. 2C and 10D). Scale bar equals 5 cm. Photograph by Christian F. Kammerer.

Figure 2 Anterior snout and dentition of Gorynychus masyutinae.

(A) Photograph and (B) interpretive drawing of the skull (KPM 346) in anterior view. (C) Disarticulated incisor (KPM 348) associated with skull in presumed anterior or anterolateral view. Abbreviations: apc, anterior premaxillary channel; mx, maxilla; na, nasal; nr, naris; pmx, premaxilla; smx, septomaxilla. Gray coloration indicates matrix. Scale bars equal 1 cm. Photographs and drawing by Christian F. Kammerer.

Figure 3 Holotype of Gorynychus masyutinae in dorsal view.

(A) Photograph and (B) interpretive drawing of skull (KPM 346). Abbreviations: d, dentary; fr, frontal; j, jugal; la, lacrimal; mx, maxilla; na, nasal; pmx, premaxilla; po, postorbital; prf, prefrontal; qpt, quadrate ramus of pterygoid; smx, septomaxilla; sq, squamosal. Gray coloration indicates matrix, patterning indicates eroded or broken bone surface. Scale bar equals 1 cm. Photograph and drawing by Christian F. Kammerer.

Figure 4 Holotype of Gorynychus masyutinae in right lateral view.

(A) Photograph and (B) interpretive drawing of skull (KPM 346). Abbreviations: ar, articular; C, upper canine; co, coronoid process of dentary; d, dentary; fr, frontal; i, lower incisor; j, jugal; la, lacrimal; mx, maxilla; na, nasal; pmx, premaxilla; prf, prefrontal; PC, upper postcanine; po, postorbital; q-qj, quadrate-quadratojugal complex; rla, reflected lamina of angular; sa, surangular; smx, septomaxilla; sq, squamosal; ss, squamosal sulcus. Gray coloration indicates matrix, patterning indicates eroded or broken bone surface. Scale bar equals 1 cm. Photograph and drawing by Christian F. Kammerer.

Figure 5 Holotype of Gorynychus masyutinae in left lateral view.

(A) Photograph and (B) interpretive drawing of skull (KPM 346). Abbreviations: C, upper canine; co, coronoid process of dentary; d, dentary; ept, epipterygoid; fr, frontal; j, jugal; la, lacrimal; mx, maxilla; na, nasal; os, orbitosphenoid; pa, parietal; pmx, premaxilla; prf, prefrontal; PC, upper postcanine; po, postorbital; qpt, quadrate ramus of pterygoid; rla, reflected lamina of angular; sa, surangular; smx, septomaxilla. Gray coloration indicates matrix, patterning indicates eroded or broken bone surface. Scale bar equals 1 cm. Photograph and drawing by Christian F. Kammerer.

Figure 6 Postcanine morphology of Gorynychus masyutinae.

Left PC1–3 in lateral view. PC2 is in the process of erupting. Scale bar equals 1 cm. Photograph by Christian F. Kammerer.

Figure 7 Holotype of Gorynychus masyutinae in ventral view.

(A) Photograph and (B) interpretive drawing of skull (KPM 346). Abbreviations: an, angular; ar, articular; bt, basal tuber; C, upper canine; d, dentary; j, jugal; mx, maxilla; oc, occipital condyle; pl, palatine; pra, prearticular; ps, parabasisphenoid; pt, palatal portion of pterygoid; q, quadrate; qpt, quadrate ramus of pterygoid; ri, rib; rla, reflected lamina of angular; sf, suborbital fenestra; sp, splenial; sq, squamosal; st, stapes; tpt, transverse process of pterygoid; v, vomer. Gray coloration indicates matrix. Scale bar equals 1 cm. Photograph and drawing by Christian F. Kammerer.

Figure 8 Cervical vertebrae of Gorynychus masyutinae (KPM 346–347).

(A) Photograph and (B) interpretive drawing. Abbreviations: as, axial neural spine; c, cervical vertebra; cr, cervical rib; ic, intercentrum; ns, neural spine; poz, postzygapophysis; prz, prezygapophysis; sf?, possible skull fragment; tp, transverse process. Gray coloration indicates matrix. Scale bar equals 5 cm. Photograph and drawing by Christian F. Kammerer.

Figure 9 Postcranial elements of Gorynychus masyutinae (KPM 347).

(A) Photograph and (B) interpretive drawing. Abbreviations: ?, unknown bone; c, cervical vertebra; cl?, possible clavicle; ri, rib; sc, scapulocoracoid; ve, vertebra. Gray coloration indicates matrix. Scale bar equals 5 cm. Photograph and drawing by Christian F. Kammerer.

Figure 10 KPM 291, a block containing disarticulated elements referred to Gorynychus masyutinae gen. et sp. nov.

(A) Photograph and (B) interpretive drawing. Abbreviations: ax, axis vertebra; d, dentary; i, incisor; po?, postorbital?; r, rib; ve, vertebra. Elements marked with asterisks are shown in greater detail in this figure. Scale bar equals 5 cm. Photograph and drawing by Christian F. Kammerer.

Figure 11 Disarticulated elements of Gorynychus masyutinae.

(A–C) are highlighted elements of KPM 291 (see Fig. 10): (A) Anterior portion of right dentary preserving lower canine; (B) ?dorsal vertebra; and (C) incisor tooth. (D) is another isolated incisor (KPM 448/1). Scale bars equal 1 cm. Photographs by Christian F. Kammerer.

LSID: urn:lsid:zoobank.org:act:105CB020-2584-4AD1-BF98-2B555EE69644

Holotype: KPM 346–349 (Figs. 1–9), a single individual (skull and cervical vertebrae in articulation, pectoral and rib elements disarticulated but directly associated with skull) broken into four pieces: KPM 346, a nearly complete skull (with damaged intertemporal region, occiput, and left temporal arcade) and lower jaws with the anterior 4 1/2 cervicals in articulation; KPM 347, postcranial elements including remaining half of fifth cervical (precise break, originally articulated with anterior portion) and worn sixth and seventh cervicals, ribs, partial clavicle, and left scapulocoracoid impression; KPM 348, isolated but associated incisor with intact crown; and KPM 349, isolated but associated incisor with damaged crown.

Referred material: KPM 291 (Figs. 10 and 11A–11C), a block of fragmentary, disarticulated elements including the anterior portion of a partial right dentary, an isolated incisor, a jugal, at least four vertebrae, several ribs, a fibula, and various indeterminate bone fragments. KPM 448/1 (Fig. 11D), an isolated incisor with damaged root.

Etymology: Named in honor of Olga Masyutina for her skillful preparation of the holotype of this taxon, as well as numerous other important specimens from the Kotelnich locality.

Diagnosis: Therocephalian distinguished from all other members of the group by its autapomorphic dental morphology: all marginal teeth serrated, with serrations forming distinct denticles that are especially prominent on the incisors and postcanines. Postcanines “spade”-shaped and reduced in number (three in the maxilla) relative to most therocephalians. Further distinguished from the other known Russian basal therocephalian Porosteognathus efremovi by a shorter tooth row on the pterygoid transverse process situated on a more discrete, raised boss and an anterolaterally-curved and expanded pterygoid palatal boss with fewer (8–9) teeth (transversely broad with ∼14 teeth in Porosteognathus).

Description

The holotype is generally well preserved (Fig. 1), with good bone quality showing surface ornamentation and sutural boundaries on most of the snout and palate (Figs. 2–7). However, the skull is somewhat crushed, the left temporal arch is broken off, and the intertemporal region is badly eroded. The anterior five cervicals are reasonably well preserved (although the atlas is not exposed as prepared), but the subsequent members of the series are badly worn (Fig. 8). Other postcranial elements are broken and worn, and the scapulocoracoid is preserved mainly as an impression (Fig. 9). The more extensive of the two referred specimens is a single block of disarticulated, fragmentary elements (Fig. 10), although most of these elements show good bone preservation (Figs. 11A–11C). One of these elements, an isolated incisor with intact crown, can confidently be referred to G. masyutinae based on the presence of very large, curved denticles on its mediodistal edges. The dentary fragment also has a tall, robust symphysis identical to that of the holotype; although it is possible this could represent a gorgonopsian, the only gorgonopsians known from this locality are much smaller and actually have weakly-developed dentary symphyses (Kammerer & Masyutin, 2018). Given the absence of any other material that is not consistent with identification as Gorynychus and the lack of overlapping elements, this set of fossils is interpreted to be the remains of a single Gorynychus individual. An additional isolated tooth (KPM 448/1; Fig. 11D) exhibits the same denticulation and general morphology as those of KPM 348 and KPM 291 (Figs. 2C and 11C) and can also be referred to G. masyutinae.

Cranium

The cranium of KPM 346 is 173 mm in standard basal length (from anteroventral edge of premaxilla to posteroventral edge of occipital condyle) and 208 mm in total dorsal length (from tip of snout to edge of temporal fenestra).

The palatal portion of the premaxilla is not exposed in the holotype because of occlusion of the lower jaw. The only information available concerning the ventral surface of the premaxilla comes from the edges of the incisor alveoli. Based on this, Gorynychus appears to have had an upper incisor count of five. Although no incisors are preserved in place, two isolated teeth (KPM 348 and 349) preserved in association with the skull have root proportions identical to the empty alveoli and are here interpreted as the upper incisors. Only one of these teeth (KPM 348) preserves an intact crown (Fig. 2C). The crown of KPM 348 is elongate, triangular, and weakly recurved with prominent mesiodistal serrations forming distinct denticles. The morphology of these denticles is unique among therocephalians: they are extremely prominent, extend from the tip of the crown right to the root, and each individual denticle is curved in the apical direction. The only comparable morphology among therocephalians occurs in the postcanines of this same specimen. The cutting edges of KPM 348 are heavily worn, but this wear is asymmetrical. As shown in Fig. 2C, one side of the tooth (it is uncertain whether this is the mesial or distal side) has more of its margin worn down than the other. The same style of asymmetrical wear, but with the sides reversed, is present on KPM 448/1 (Fig. 11D). The isolated incisor of KPM 291, by contrast, is relatively unworn (Fig. 11C). It is uncertain whether this is simply due to recent eruption before the death of the animal or from occupying a different position in the tooth row (its crown is shorter and slightly more curved than in KPM 348 or KPM 448/1, which could be because it represents a more posterior tooth position or a lower incisor).

The facial surface of the premaxilla is overlain by the maxilla posteriorly, near the point between the alveoli for I4 and I5 (Figs. 2, 4 and 5). The bone surface of the premaxilla is rugose, with a series of well-developed foramina located 3–4 mm above the alveolar margin. Anteriorly, three foramina are situated in a distinct channel that originates vertically beneath the internarial bar then curves posterolaterally (Fig. 2B). The first and third foramina in this channel are small (<1 mm diameter), but the second is large (1 mm diameter) and situated deep inside the groove. Posterior to this channel, several additional foramina are present on the lateral face of the premaxilla. The internarial bar is angled somewhat anteriorly, such that it overhangs the alveolar portion of the premaxilla in lateral view (Fig. 4). The ascending ramus of the premaxilla frames the anterodorsal border of the external naris and terminates near the posterodorsal narial border.

The septomaxilla consists of a broad ventral plate making up the base of the external naris, a constricted intranarial portion, and a facial process extending between the maxilla and nasal (Figs. 2, 4 and 5). The ventral plate of the septomaxilla is situated immediately dorsal to the premaxillary–maxillary suture. Immediately ventral to this plate is a large, ovoid foramen (1.5 mm diameter) that spans the premaxillary–maxillary suture. A weak groove extends anterior to this foramen for the length of the ventral plate of the septomaxilla. The constricted intranarial portion of the septomaxilla separates the main portion of the external naris from the maxillo-septomaxillary foramen. It has a pointed, anteromedially-directed anterior process as is typical of therocephalians, but not an expanded transverse lamina as in gorgonopsians. Posterodorsally, the septomaxilla makes a broad, irregular contribution to the facial portion of the snout. This facial process notably does not make up the entire posterior margin of the external naris—the nasal occupies the posterodorsal narial corner (Fig. 2B).

The maxilla is a tall, gently curved bone making up the main lateral surface of the snout (Figs. 4 and 5). The facial surface of the maxilla is intensely sculptured with shallow, crater-like structures and small foramina, similar to that of some other basal therocephalians (see, e.g., Abdala, Rubidge & van den Heever, 2008). A labial emargination is absent, unlike in Lycosuchus. The posterior terminus of the maxilla is a process under the jugal that reaches the midpoint of the orbit. The tip of this process is notably expanded and rugose, but the majority of this process is notable for lacking the dermal sculpturing so prominent on the rest of the maxilla and bearing a weak lateral groove. The maxilla houses four teeth: the enlarged, blade-like canine and three small postcanines. The canine is recurved and has well-developed serrations fore and aft, as is usual in basal therocephalians (van den Heever, 1994; Abdala et al., 2014). Only a single canine is erupted on each side; unlike the usual condition in lycosuchids and a frequent condition in scylacosaurids, there are not simultaneously-occupied anterior and posterior canine alveoli. The tip of a replacement canine is erupting anterior to the right canine, however. The postcanine morphology of Gorynychus is unique among therocephalians—these teeth are labiolingually compressed and “leaf” or “spade”-shaped, with proportionally large denticles (serrations) along both mesial and distal edges (Fig. 6). Although these denticles are not as large or exaggeratedly curved as those on the incisors, they still show distinct curvature (unlike the finer, straight serrations on the canine). As for the premaxilla, the palatal surface of the maxilla is for the most part not visible in this specimen.

The nasal is a long, flat bone (Figs. 2–5). It also bears dermal sculpturing, albeit developed to a lesser degree than in the facial portion of the maxilla. Dermal sculpturing to some degree is present on all the bones of the snout, including (albeit very weakly) on the facial portion of the jugal. The naso-frontal suture is slightly bowed posteriorly and terminates near the anterior margin of the orbit.

The prefrontal is a roughly trapezoidal bone making up part of the snout and the anterodorsal margin of the orbit (Figs. 3–5). Its contribution to the orbital rim is unusually pronounced and rugose. This is also true for the lacrimal, jugal, and frontal, such that that the circumorbital rim is very prominent everywhere but on the postorbital bar. Anteriorly, the prefrontal terminates in a short process extending between the maxilla and nasal. The prefrontal makes up a large portion (roughly half) of the anterior face of the orbital wall, and appears to exclude the frontal from contacting the lacrimal.

The lacrimal is a large facial bone, nearly equivalent in height to the prefrontal albeit not as anteroposteriorly long (Figs. 4 and 5). A similarly-sized lacrimal is also present in Lycosuchus, and although this bone is less dorsoventrally expanded in scylacosaurids it is of similar anteroposterior dimensions in that group as well (van den Heever, 1994). A large (1.5 mm diameter) lacrimal foramen is present on its contribution to the anterior orbital wall; this foramen does not exit onto the lateral surface. A second, smaller lacrimal foramen may be present ventral to the first, but this is uncertain because of damage.

The jugal makes up the majority of the zygomatic arch (Fig. 4). Its facial portion is an attenuate process terminating below the lacrimal, posterior to the anterior lacrimal margin. This facial portion is remarkably small for an early theriodont. In gorgonopsians, the jugal typically terminates in a broad, plate-like contribution to the snout equalling (or exceeding) the lacrimal in anterior extent (see, e.g., Kammerer, 2015; Kammerer et al., 2015). In Lycosuchus and other early therocephalians, the facial portion of the jugal terminates well posterior to the anterior edge of the lacrimal, but still forms a tall plate on the snout and occupies almost the entire ventral margin of the orbit (van den Heever, 1994). The jugal is constricted beneath the posterior margin of the orbit, before expanding to form a major part of the postorbital bar. In the postorbital bar, an ascending process of the jugal extends dorsally behind the postorbital. Posterior to the postorbital bar the jugal makes a tall contribution to the subtemporal bar, narrowing posteriorly and curving downwards. In the posterior half of the subtemporal bar, the jugal is bifurcated by an anterior process of the squamosal. The ventral portion of the jugal extends nearly to the back of the skull, and has a weakly bulbous terminus. In addition to being preserved in the holotype, an isolated jugal is present on the referred block (Fig. 10), identifiable by the characteristic subtemporal curvature and well-developed process contributing to the postorbital bar.

The squamosal is preserved mainly in its zygomatic ramus (Fig. 4), as the occiput and intertemporal region are poorly preserved in this specimen (Fig. 1). It has a deep and fairly restricted squamosal sulcus, giving it a “forked” appearance in lateral view. Medially, it forms the posterior portion of the lateral margin of the temporal fenestra (anteriorly, this margin is formed by the jugal).

The frontal is a mostly-flat bone making up the interorbital skull roof (Fig. 3). The medial portion of the frontal is damaged in this specimen, but its contribution to the orbit is well preserved and shows that it was only weakly ornamented dorsally, until the very edge of the orbit (in which it is expanded and rugose, like most of the circumorbital rim). The suture between the frontal and postorbital is poorly preserved and difficult to interpret, but it appears that the frontal makes only a narrow contribution to the dorsal orbital wall (Fig. 4), as in Lycosuchus (van den Heever, 1994).

A distinct postfrontal is absent in Gorynychus; this element may be fused with the postorbital. The postorbital makes up a broad portion of the skull roof at the posterodorsal margin of the orbit and continues as a posterior ramus making up the medial face of the temporal fenestra (Figs. 3 and 4). The latter ramus is badly damaged in this specimen, like most of the posterior skull roof, and its posterior terminus and surface texture is indeterminable. The intertemporal bar is narrow, as is typical of therocephalians. However, this region is too damaged to determine its exact proportions or presence/absence of the pineal foramen or sagittal crest. The occiput is even more badly damaged, being worn off entirely dorsal to the occipital condyle (Fig. 5).

The vomer is not exposed anteriorly, but its posterior interchoanal portion is a tall, narrow blade, like that of scylacosaurids and unlike the broad, vaulted morphology of Lycosuchus (Fig. 7). The vomer is paired—a distinct mid-vomerine suture is visible in the interchoanal portion. Posteriorly, the vomer forms a broad, triangular contribution to the palate, extending between the palatines to contact the anterior margin of the pterygoids posteriorly. An elongate palatal depression or fossa is present medially, originating near the anterior edge of the plate-like palatal portion of the vomer and extending posteriorly until it reaches the transverse processes of the pterygoids. Although the depth of this structure is likely exaggerated by lateral compression in this specimen, its presence is natural.

The palatine is the largest bone of the palate (Fig. 7). It is a topologically complex element composed of a laminar anterior process that forms much of the lateral margin of the choana and a broad main portion with a prominent central ridge bounded by medial and lateral depressions. This central ridge extends anterolaterally to posteromedially and is confluent posteriorly with a ridge on the pterygoid, terminating with the palatal boss. A suborbital fenestra, characteristic of therocephalians, is present at its posterolateral margin, bounded posteriorly by the pterygoid. Presumably it is also bounded laterally by the ectopterygoid, but this element is indistinct if present, probably due to poor preservation of the lateral margins of the palate.

As is typical of early therapsids, the pterygoid consists of three distinct processes: palatal, transverse, and quadrate (Fig. 7). The palatal portion of the pterygoid has an anteromedial-to-posteromedially-angled, strongly interdigitated suture with the palatine. Anteriorly it forms a short portion of the prominent palatal ridges (central ridge of palatine) before expanding into the dentigerous palatal bosses posteriorly. Dentigerous palatal bosses are ancestral for therapsids; although they are present in scylacosaurids and even various eutherocephalians (van den Heever, 1994; Huttenlocker & Smith, 2017), they are absent in lycosuchids (Lycosuchus and Simorhinella) (Abdala et al., 2014). The palatal boss is “teardrop”-shaped, with a narrow posterior tip and broad, rounded anterior. It is curved anterolaterally. The boss bears two tooth rows. On the left palatal boss (the more completely preserved of the two), the posterior tooth row follows the curve of the boss and is made up of five teeth. The anterior tooth row is transversely oriented and is made up of four teeth, for a total of nine (six are present on the right boss, but as mentioned this boss is damaged and this is probably not the complete complement of teeth). The transverse process of the pterygoid extends ventrolaterally: it has a broad base medially, constricts in ventral view toward its midpoint, and then expands into a rounded lateral tip where it braces the mandible. The anterior face of the transverse process bears a broad, shallow depression. Unlike most therocephalians, but similar to lycosuchids, the transverse process is dentigerous in Gorynychus. Unlike Lycosuchus and Simorhinella, however, where the tooth row is relatively long (five teeth) and the teeth erupt directly from the main ramus of the transverse process, in Gorynychus there are only 2–3 teeth situated on discrete, ovoid bosses raised above the medial bases of the transverse processes. On the right transverse process there are only two teeth, a large lateral and small medial one; on the left there are three teeth of equal size, similar to the small one on the right. No interpterygoid vacuity is present; although there is a weak depression between the transverse processes medially, it is entirely bounded by bone dorsally. The quadrate process of the pterygoid is situated dorsal to the transverse process. It extends posterolaterally from a position near the medial base of the transverse process toward the quadrate, weakly curving along its length. The anterolateral margin of the quadrate process forms a tall, narrow ridge; posteromedially it forms a broad, concave plate bounding the lateral edge of the parasphenoid rostrum.

The parasphenoid-basisphenoid complex forms a narrow median rostrum originating behind the transverse processes of the pterygoids and extending posteriorly to the basal tubera (Fig. 7). Although tall and blade-like posteriorly, the anterior two-thirds of this complex are divided by a narrow median groove. The posterior terminus of the rostrum is abrupt, with a sharp drop to the base of the basal tubera (typical of early therocephalians) instead of a gradual decrease in height. Dorsally, the parasphenoid forms a narrow median lamina above the pterygoid, forming part of the mid-orbital plate (Fig. 5). Dorsal to this anteriorly is a tall, laminar bone interpreted as the orbitosphenoid, which extends dorsally to contact the frontal-postorbital wall of the orbit. Dorsal to it posteriorly is the epipterygoid, which has an anteroposteriorly broad footplate ventrally that sits atop the pterygoid. The ascending process of the epipterygoid narrows dorsally before expanding again at its contact with the ventral face of the parietal. The basal tubera are broadly separated by a median depression (Fig. 7). They are relatively slender and angled medially at their posterior end. The left stapes is preserved in place, extending from the basal tuber to near the quadrate. The stapes is dorsoventrally narrow but anteroposteriorly broad, although narrowing along its length laterally (similar to Lycosuchus). No foramen or dorsal process is visible, but these could be obscured by matrix. The basioccipital forms a plate posterior to the basal tubera and terminates in the occipital condyle, which is similar to that of other therocephalians.

The dentary is a massive, robust bone with a tall, well-developed, unfused symphysis more similar to that of gorgonopsians than other early therocephalians (Figs. 2, 4, 5 and 11A). The anterior face of the symphysis is densely foraminated. The roots of at least three incisors are exposed due to damage to the right dentary (Fig. 4), but it is probable that more were present, given that these three do not occupy the entirety of the symphysial length. The crowns of the lower incisors are not exposed. The lower canine is not exposed in the holotype but is well-preserved in the disarticulated dentary fragment in KPM 291 (Figs. 10 and 11A). The lower canine is a tall, recurved tooth with well-developed fore and aft serrations. It is proportionally large, taking up much of the alveolar margin of the symphysis. Based on this position, it would have been situated anterior to the upper canine when in occlusion with the cranium. The lower postcanines are mostly obscured in both known specimens of Gorynychus, only a single small lower postcanine is exposed on the right mandibular ramus anterior to the upper PC1 in the holotype. This postcanine is smaller than any of the upper postcanines (2 mm apicobasal length, versus 4 mm in the uppers) but is otherwise similar in morphology, being “spade”-shaped with well-developed denticulation. The dentary is constricted behind the symphysis, then expands posteriorly, with a well-developed angular process (Fig. 4). The dorsal and ventral margins have raised edges, and a distinct lateral fossa is present between them, extending anteriorly almost to the level of the symphysis. Although superficially similar to the masseteric fossa of cynodonts, this fossa is likely non-homologous; there is no evidence that the superficial masseter was present in therocephalians. The raised ventral margin of the dentary terminates posteriorly in a broad, flattened rugose region serving as the attachment site for adductor musculature. The coronoid process extends freely above the postdentary bones and has a broad posterodorsal terminus, as in the other basal therocephalians (van den Heever, 1994). Unlike Lycosuchus, however, in which the posterior edge of the coronoid process is broadly rounded, in Gorynychus this edge is nearly straight (slightly concave).

The postdentary bones are damaged on both sides of the skull: the surangular and articular are more complete on the right side but the reflected lamina of the angular is broken off ventrally; the surangular and posterior portion of the angular are broken off on the left side but the reflected lamina is more complete (Figs. 4 and 5). In general, the postdentary elements are very similar to those of lycosuchids and scylacosaurids. The reflected lamina is large and occupies the entire lateral surface of the angular (typical of early therapsids, but distinct from gorgonopsians in which it is usually widely separated from the articular). The surface structure of the reflected lamina is typical of early therocephalians: an anterodorsal depression becoming a single broad ridge posteroventrally that then ramifies into ventral and posterior ridges. The surangular is exposed laterally as a narrow, curved element atop the angular, contacting the articular posteroventrally. A short angle is present at the posterior base of the coronoid process of the dentary where it overlies the surangular, and the dentary is weakly raised laterally anterior to this point, accommodating the internal anterior process of the surangular. The splenial is a tall, “ribbon”-like bone occupying the medial face of the anterior portion of the jaw ramus (Fig. 7). It lacks any lateral exposure, being restricted to the internal surface of the jaw. It covers almost all of the medial surface of the dentary anteriorly, but decreases in height posteriorly before terminating at the level of the transverse process of the pterygoid. Dorsal to the splenial is a narrow, laminar bone: the prearticular. This element broadens posteriorly and becomes more raised and rod-like posterior to the dentary, terminating in an indistinct contact with the articular. Dorsal to the prearticular at roughly the mid-length of the jaw ramus is a short, narrow, laminar element interpreted as being the coronoid. The articular is poorly preserved on both sides in this specimen, but a bulbous, cup-like terminus articulating with the (equally poorly preserved) quadrate can be discerned in medial view on the left side (Figs. 4 and 7). Although mostly worn off, the base of a large retroarticular process is present.

Postcranium

Little of the postcranium is preserved in the holotype of G. masyutinae. The majority of the cervical series is preserved in articulation with the skull and some ribs and pectoral elements are preserved posterior to this (Fig. 1). The atlas-axis complex is somewhat damaged and obscured by the skull and surrounding matrix and C5–7 are broken and badly worn, but C3 and 4 are well-preserved and exposed on both sides (Fig. 8). The cervical vertebrae are amphicoelous (except, presumably, the atlas) and are separated ventrally by small, wedge-shaped intercentra. The transverse processes are short and blunt. At least one cervical rib is preserved on the right side of the specimen in association with (but disarticulated from) vertebra C3. This rib has a broadly falcate head and mediolaterally narrow main body. Well-developed, stout prezygapophyses (12 mm anteroposterior length) extending anterior to the centra are present in C3 and 4, contacting the associated postzygaphophyses of the anterior vertebrae ventrally (and slightly laterally, as the prezygapophysis bulges out somewhat at its anterodorsal edge). No anapophyses are present. The neural spines of C3 (17 mm tall) and 4 are constricted immediately above their point of origin but expand dorsally into broad, rounded tips (anteroposteriorly 11 mm long in C3). The axial neural spine is definitely anteroposteriorly broader than that of the subsequent vertebrae (Fig. 8), but its exact dimensions (including height relative to subsequent neural spines) are uncertain due to crushing. An isolated vertebra interpreted as the axis in the referred specimen KPM 291 preserves a tall, broad neural spine (Fig. 10), suggesting that the axial spine was substantially taller and longer than subsequent neural spines, as is typical for theriodonts (Jenkins, 1971). The other vertebrae preserved in KPM 291 have relatively lower, longer centra and shorter neural spines (Fig. 11B) than those of C3 and 4 in the holotype. They likely represent dorsals or even anterior caudals, although most are too poorly preserved to identify with any confidence.

The non-vertebral postcranial elements in the holotype are mostly damaged (Fig. 9) The remains of at least eight ribs are present, but little of their morphology is exposed other than simple, curved shafts. A robust, curved elongate bone exposed in worn cross-section probably represents the clavicle, but nothing more about its morphology can be said. The left scapulocoracoid is preserved almost entirely as impression (there is a small chunk of actual bone from the anterior margin of the procoracoid). This impression shows that the coracoid-procoracoid base of the structure was very anteroposteriorly long. The scapula is broad ventrally but narrows markedly dorsally, where it curves anteriorly at tip. A prominent ridge on the scapula originates at the posterior margin of this element before curving anteriorly along the length of the scapular spine. An elongate bone preserved as part of KPM 291 (Fig. 10) is here identified as a fibula. This element lacks the curvature seen in the cervical-thoracic ribs on this block and in KPM 347, and is too long to be a lumbar rib. Additionally, it is expanded at both ends, unlike a rib. The morphology of this element is similar to the fibula of other therocephalians, in which it is usually a narrow, simple bone (Fourie & Rubidge, 2009).

Phylogenetic Analysis

Gorynychus masyutinae was coded into a recent analysis of therocephalian interrelationships, that of Huttenlocker & Smith (2017). This analysis (available as Supplemental Information) includes 136 discrete-state characters and 58 taxa, including two newly added to the analysis: G. masyutinae, coded based on personal examination of all known specimens, and Shiguaignathus wangi, coded based on the figures in Liu & Abdala (2017). Analysis was undertaken using heuristic searching in PAUP* (Swofford, 2002) v.4.0a (build 159), treating Biarmosuchus tener as the outgroup. All characters were treated as unordered following Huttenlocker & Smith (2017). Bootstrap analysis was done using “fast” stepwise addition for 1,000 replicates.

A total of 1,260 most parsimonious trees of length 383 were recovered (consistency index = 0.420, retention index = 0.785). The strict consensus tree recovers G. masyutinae as the sister-taxon of Eutherocephalia (i.e., Akidnognathidae + Whaitsioidea + Baurioidea) (Fig. 12). A position outside of Eutherocephalia is supported by the presence of a paired vomer, serrations on all marginal teeth, and teeth on the transverse process of the pterygoid in Gorynychus (all of which are typically absent in eutherocephalians) and the absence of a well-developed mandibular fenestra penetrating the jaw in lateral view. The absence of the postfrontal in Gorynychus is a character shared with eutherocephalians to the exclusion of lycosuchids and scylacosaurids (although this element is still variably present in some hofmeyriids). The addition of Gorynychus adds substantial instability to the analysis of Huttenlocker & Smith (2017), notably with the complete collapse of Whaitsioidea (although a core Hofmeyriidae composed of Hofmeyria, Ictidostoma, and Mirotenthes is retained). Another Kotelnich taxon, Perplexisaurus, also falls outside of Eutherocephalia in the current analysis, occupying the node between Scylacosauridae and (Gorynychus + Eutherocephalia). In previous analyses, Perplexisaurus was a highly unstable taxon generally occupying a position somewhere near the base of Eutherocephalia (Huttenlocker & Sidor, 2016; Huttenlocker & Smith, 2017), so its continued lability is not especially surprising.

Figure 12 Cladogram showing phylogenetic position of Gorynychus masyutinae.

Strict consensus of 1,260 most parsimonious trees. Numbers at nodes represent bootstrap values, major clades labeled at nodes. Image by Christian F. Kammerer.

The breakdown of one of the major eutherocephalian clades (Whaitsioidea) as the sole result of adding Gorynychus to the dataset indicates remarkably poor support for a group that, on a strictly gestalt basis, seems to be very well-characterized. This problem can likely be blamed on extensive homoplasy in Eutherocephalia, the mosaic of features present in Gorynychus, as well as the need for additional sources of phylogenetic data. Existing phylogenetic data sets for Therocephalia are heavily skewed toward craniodental characters; for example, only 19/136 characters in Huttenlocker & Smith’s (2017) analysis are postcranial. Although cranial-focused analyses are typical for Therapsida (an artifact, in part, of preferential collection of skulls by Karoo paleontologists during most of the 20th century), recent analyses of synapsid relationships (e.g., Benson, 2012) have highlighted the importance of bringing more robust sets of postcranial data to bear on phylogenetic problems in this clade. Before such data can be incorporated into therocephalian analyses, however, more basic descriptive work on therocephalian postcrania is needed. Although skeletons are now known for a wide array of taxa, anatomical descriptions are currently available for only a select few (Kemp, 1986; Fourie & Rubidge, 2009; Botha-Brink & Modesto, 2011; Fourie, 2013).

Discussion

At roughly 20 cm in skull length, Gorynychus is the largest predatory component of the Kotelnich tetrapod assemblage. Co-occurring gorgonopsians are substantially smaller, with skull lengths less than 10 cm (Kammerer & Masyutin, 2018). The only other Kotelnich predators approaching Gorynychus in size are the eutherocephalian Viatkosuchus (co-occurring with Gorynychus in the Vanyushonki Member) and the burnetiamorph Proburnetia (known only from the holotypic mold from the younger Sokol’ya Gora site) (Ivakhnenko, 2011; Benton et al., 2012). This situations parallels that of middle–earliest late Permian faunas in South Africa (the Tapinocephalus and Pristerognathus AZs), wherein the only gorgonopsians are small animals and therocephalians are the larger-bodied predators (Kammerer, 2014). The Kotelnich predatory fauna is particularly comparable to that of the Pristerognathus AZ; in the middle Permian Tapinocephalus AZ, although large-bodied therocephalians were abundant, the apex predators were gigantic anteosaurian dinocephalians. The extinction of anteosaurs at the end of the Capitanian left a depauperate fauna with therocephalians momentarily atop the food chain (Day et al., 2015). It was only following the later extinction of these basal therocephalians that gorgonopsians began to diversify and cemented their position as the dominant large-bodied therapsid predators (Kammerer et al., 2015).

Although the presence of a therocephalian apex predator and relatively small gorgonopsians in Kotelnich indicates that the earliest South African records may accurately reflect ancestral body size in these clades (rather than regional peculiarities), in other regards the Kotelnich fauna differs markedly from that of the Karoo. The dominant herbivorous taxon in the Kotelnich fauna, and most abundant tetrapod of any kind, is the pareiasaur Deltavjatia (Tsuji, 2013). This pattern is in stark contrast to that of South Africa, where dicynodont therapsids are numerically dominant in all middle–late Permian faunas (Smith, Rubidge & van der Walt, 2012). Furthermore, this does not seem to be an isolated oddity, as the best-sampled later Permian Russian tetrapod fauna (North Dvina, a.k.a. Sokolki) also has pareiasaurs (Scutosaurus) as the dominant component (although there the top predator is the giant gorgonopsian Inostrancevia; this inferred predator-prey pair represents the “oligobiomorph community” of Ivakhnenko (2008)).

The therocephalian fauna in Kotelnich is also unusual even by comparison to the Pristerognathus AZ. Although eutherocephalian fossils have been found in the Pristerognathus AZ, they are rarer than scylacosaurids/lycosuchids and are relatively poorly known (Huttenlocker & Smith, 2017). By contrast, eutherocephalians are the most abundant and species-rich theriodonts at Kotelnich. As such, the Kotelnich fauna seems to capture a phase in therocephalian evolution not well-represented in the South African record: the initial diversification of Eutherocephalia. The recovery of Gorynychus as a taxon just outside of Eutherocephalia, rather than a lycosuchid or scylacosaurid, adds another wrinkle to this interpretation. The few eutherocephalians known from the Pristerognathus AZ belong to well-known groups (the whaitsioid family Hofmeyriidae and baurioid family Ictidosuchidae) deeply nested within Eutherocephalia and appear in the record seemingly without precedent. In Kotelnich, however, there is a combination of taxa at the base of Eutherocephalia (Gorynychus, possibly Perplexisaurus) and potential early representatives of known eutherocephalian families (e.g., Viatkosuchus). This indicates substantial therocephalian diversification occurring in Laurasia at the time, and suggests that eutherocephalians originated outside of the Karoo and only later migrated to the basin.

Conclusion

Based on a nearly-complete skull and partial skeleton and two additional, fragmentary specimens, a new therocephalian taxon, G. masyutinae, is described from the (probably) earliest late Permian Kotelnich locality of Russia. Gorynychus is the largest known predatory tetrapod in the Kotelnich assemblage, and demonstrates that therocephalians acted as top predators in Russian as well as South African assemblages during the transition between typical middle and late Permian terrestrial communities. Although falling outside of Eutherocephalia, Gorynychus is more closely related to eutherocephalians than to the large-bodied therocephalian predators of southern Africa (and possibly earlier Permian assemblages in Russia, if Porosteognathus from the middle Permian Isheevo fauna truly is a scylacosaurid). The Kotelnich therocephalian fauna shows greater diversity of eutherocephalians than probable coeval faunas in South Africa, and suggests that initial diversification in this clade probably was not occurring in the Karoo Basin.

Supplemental Information

Supplemental Information 1 Character matrix for phylogenetic analysis.

Click here for additional data file.

We thank director A. Toporov, curator T. Berestova, and the other staff members at the Vyatka Paleontological Museum for all of their help. We thank Fernando Abdala and Jun Liu for their helpful reviews of the original manuscript.

Institutional abbreviation

KPM Vyatka Paleontological Museum, Kirov, Russia.

Additional Information and Declarations

Competing Interests

Author Contributions

Data Availability

New Species Registration

The authors declare that they have no competing interests.

Christian F. Kammerer conceived and designed the experiments, performed the experiments, analyzed the data, contributed reagents/materials/analysis tools, prepared figures and/or tables, authored or reviewed drafts of the paper, approved the final draft.

Vladimir Masyutin contributed reagents/materials/analysis tools, authored or reviewed drafts of the paper.

The following information was supplied regarding data availability:

The character matrix used in the phylogenetic analysis is provided as a Supplemental File.

The following information was supplied regarding the registration of a newly described species:

Publication LSID: urn:lsid:zoobank.org:pub:CA4D73A1-8FA7-40DD-A464-621AC01421B6;

Gorynychus: urn:lsid:zoobank.org:act:CD10EB7C-57C0-45BA-8467-75309411E0DD;

Gorynychus masyutinae: urn:lsid:zoobank.org:act:105CB020-2584-4AD1-BF98-2B555EE69644.

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
