# Peer review of "A new therocephalian (Gorynychus masyutinae gen. et sp. nov.) from the Permian Kotelnich locality, Kirov Region, Russia"

_PeerJ, doi:10.7717/peerj.4933_

## Round 0.1 · original submission · Minor Revisions

Please respond to the reviewers' concerns point by point when preparing a revised version of the manuscript. Please check the data matrix as one referee could not access it.

·

Basic reporting

The original matrix file cannot be read directly by Paup, please change it.

Experimental design

The phylogenetic analysis should include all taxa such as Shiguangnathus.

Validity of the findings

This is a new therocephalian species, and it is correctly described.
I am not convinced it is the sister-taxon to Eutherocephalia.
The author did not mention Scylacosuchus moved as the sister taxon of Akidognathidae. This needs to be discussed. If they coded the new species in the matrix of Liu and Abdala (2017), the result is slightly different: Gorynychus is more basal than Scylacosauridae in most cases. There should be a paragraph discussing the position of the new species based on the morphology, other than just the phylogenetic analysis.
One error is on postfrontal, the author described it but later say it is absent.

Additional comments

Other see annotated pdf file.

·

Basic reporting

The figures are good but I would like to see close up of the postcanines as it is indicated as one important feature in the description. Other suggestions are in the uploaded pdf.

Experimental design

No comments

Validity of the findings

In the discussion it is assumed a correlation with the Pristerognathus AZ, even when in the introduction it is said that the correlation with the Tropidostoma AZ seems more likely. Considering one or the other case change the conclussion about the timing in the evolutionof Eutherocephalia. Thus if the correlation is with the Pristerognathus AZ then the conclusion about the early evolution of the group in Laurasia is supported but if the correlation is with Tropidostoma AZ then this conclussion will change or woud be less strong.

Additional comments

This is a very important research of a new therocephalian from the Kotelnich fauna from Russia. It is clearly written with a clear and logic structure, presenting in an apropriate way the subject. I present all my comments and suggestions in the attached pdf. The article should be published after minor revision.

---

## Round 0.2 · accepted · Accept

The revised version of the manuscript is acceptable for publication.

#